# BST204 Protects Dexamethasone-Induced Myotube Atrophy through the Upregulation of Myotube Formation and Mitochondrial Function

**DOI:** 10.3390/ijerph18052367

**Published:** 2021-03-01

**Authors:** Ryuni Kim, Hyebeen Kim, Minju Im, Sun Kyu Park, Hae Jung Han, Subin An, Jong-Sun Kang, Sang-Jin Lee, Gyu-Un Bae

**Affiliations:** 1Research Institute of Pharmaceutical Science, College of Pharmacy, Sookmyung Women’s University, Seoul 04310, Korea; rty0424@naver.com; 2Molecular Cell Biology, Single Cell Network Research Center, School of Medicine, Sungkyunkwan University, Suwon 16419, Korea; bin3981@hanmail.net (H.K.); thdrlwk02@hanmail.net (S.A.); kangj01@skku.edu (J.-S.K.); 3Green Cross Wellbeing Co., Ltd., Seongnam 13595, Korea; minjulim@greencross.com (M.I.); plantist@greencross.com (S.K.P.); hjhan@greencross.com (H.J.H.); 4Research Institute of Aging-Related Disease, AniMusCure Inc., Suwon 16419, Korea

**Keywords:** Akt/mTOR pathway, BST204, mitochondria, muscle atrophy, PGC1α

## Abstract

BST204 is a purified ginseng dry extract that has an inhibitory effect on lipopolysaccharide-induced inflammatory responses, but its effect on muscle atrophy is yet to be investigated. In this study, C2C12 myoblasts were induced to differentiate for three days followed by the treatment of dexamethasone (DEX), a corticosteroid drug, with vehicle or BST204 for one day and subjected to immunoblotting, immunocytochemistry, qRT-PCR and biochemical analysis for mitochondrial function. BST204 alleviates the myotube atrophic effect mediated by DEX via the activation of protein kinase B/mammalian target of rapamycin (Akt/mTOR) signaling. Through this pathway, BST204 suppresses the expression of muscle-specific E3 ubiquitin ligases contributing to the enhanced myotube formation and enlarged myotube diameter in DEX-treated myotubes. In addition, BST204 treatment significantly decreases the mitochondrial reactive oxygen species production in DEX-treated myotubes. Furthermore, BST204 improves mitochondrial function by upregulating the expression of peroxisome proliferator-activated receptor-γ coactivator-1α (PGC1α) in DEX-induced myotube atrophy. This study provides a mechanistic insight into the effect of BST204 on DEX-induced myotube atrophy, suggesting that BST204 has protective effects against the toxicity of a corticosteroid drug in muscle and promising potential as a nutraceutical remedy for the treatment of muscle weakness and atrophy.

## 1. Introduction

The progressive loss of skeletal muscle mass and strength is a hallmark of muscle aging, leading to reduced functional capacity and an increased risk of developing chronic metabolic diseases [1,2]. Muscle atrophy is triggered by an imbalance between protein synthesis and protein degradation by the ubiquitin-proteasome system (UPS), resulting in a decrease in muscle fiber size [3]. Muscle RING finger containing protein 1 (MuRF1) and Atrogin-1 are major muscle-specific E3 ubiquitin ligases associated with protein degradation in muscle cells [4,5] and are high-fidelity markers of the muscle atrophy phenotype [6]. In addition to the imbalance in protein metabolism, the decreased regenerative capacity of muscle stem cells is also tightly associated with the loss of muscle mass and function resulting from aging or age-related muscle diseases [3,7]. Akt (protein kinase B) is a well-known promyogenic signaling in myoblast differentiation [8,9,10], and plays a critical role in insulin-like growth factor-mediated myoblast proliferation and survival [11,12]. Akt/mTOR (mammalian target of rapamycin) signaling is also associated with protein synthesis and muscle hypertrophy. Akt/mTOR signaling inhibits the transcriptional function of Forkhead box O (FoxO), which induces the transcriptional upregulation of MuRF1 and Atrogin-1 [13,14].

BST204 is purified ginseng dry extract from the root of *Panax ginseng* C.A. Meyer. It is derived from crude ginseng using ginsenoside-β-glucosidase and acid hydrolysis, to enrich the ginsenosides Rg3 and Rh2 [15]. BST204 has been reported to have inhibitory effects on lipopolysaccharide-induced inflammation, through downregulation of the expression of inducible nitric oxide synthase and cyclooxygenase-2 [15,16]. BST204 ameliorates cancer-related fatigue by regulating inflammatory responses and hematopoiesis [17]. However, the underlying molecular mechanism of BST204 in myotube atrophy has not been investigated so far. 

Recent advances in muscle biology, especially the understanding of the molecular regulation of muscle mass and function by metabolic incentives, have aroused new interest in the use of pharmacological therapeutics to prevent muscle atrophy. In the present study, we investigated the molecular mechanism behind the effect of BST204 on dexamethasone (DEX)-induced myotube atrophy. We found that BST204 improves myotube formation by activating Akt/mTOR signaling in DEX-treated myotubes and enhances PGC1α-mediated mitochondrial function to prevent DEX-triggered myotube atrophy. Thus, BST204 has protective effects against the toxicity of a corticosteroid drug in muscle and might be a potential nutraceutical for the treatment of muscle weakness and atrophy.

## 2. Materials and Methods

### 2.1. Cell Culture

C2C12 cells were cultured as described previously [18]. To induce differentiation of C2C12 myoblasts, cells at near confluence were switched from DMEM (Dulbecco modified Eagle’s medium, Thermo Fisher Scientific, Waltham, MA, USA.) containing 15% FBS (fetal bovine serum, Thermo Fisher Scientific; growth medium (GM)) to DMEM containing 2% HS (horse serum, Thermo Fisher Scientific; differentiation medium (DM)), and myotube formation was observed normally at approximately 2–3 days of differentiation. The efficiency of myotube formation was quantified by a transient differentiation assay, as previously described [19]. For the myotube atrophy study induced by DEX (Sigma-Aldrich, St. Louis, MO, USA), C2C12 cells were induced to differentiate in DM for 3 days and treated with 10 μM DEX and the indicated concentration of BST204, followed by incubation in DM for 1 day [20].

### 2.2. Western Blotting Analysis, Immunocytochemistry, and Microscopy

Western blotting analysis was carried out as previously described [18,19]. Briefly, cells were lysed in extraction buffer (50 mM Tris-HCl, pH 7.4, 150 mM NaCl, 1.2 mM MgCl_2_, 1 mM EGTA, 1% Triton X-100, 10 mM NaF, and 1 mM Na_3_VO_4_) containing complete protease inhibitor cocktail (Roche Diagnostics, Basel, Switzerland). Sodium dodecyl sulfate-polyacrylamide gel electrophoresis (SDS-PAGE) was performed, followed by immunoblotting. The primary antibodies used in this study were as follows: MHC (MF-20; Developmental Studies Hybridoma Bank, Iowa City, IA, USA), phospho-Akt, Akt, phospho-mTOR, mTOR, phospho-p70S6K, p70S6K, PGC1α, NRF1, Tfam (Cell Signaling Technology, Beverly, MA, USA), myogenin, α-tubulin (Santa Cruz Biotechnology, Santa Cruz, CA, USA), MuRF1 (GeneTex, Irvine, CA, USA), Atrogin-1 (ECM Biosciences, Versailles, KY, USA), and pan-Cadherin (Sigma-Aldrich, St. Louis, MO, USA).

Immunostaining for MHC expression was performed as described previously [19]. Briefly, the differentiated cultures were immunostained for MHC antibodies and Alexa 568-conjugated secondary antibodies (Molecular Probes). Images were captured and processed with a Nikon ECLIPSE TE-2000U microscope and NIS-Elements F software (Nikon, Tokyo, Japan). MHC-positive myotubes with 10 and more nuclei were measured transverse diameter in six fields. Quantification of myotube diameter was performed with Image J software (Fujifilm, Tokyo, Japan). Average myotube diameter is presented as means determination of six fields ± 1 standard deviation (SD). 

### 2.3. RNA Extraction and Quantitative RT-PCR (qRT-PCR)

Total RNA extraction and qRT-PCR analysis were carried out as described previously [21]. Briefly, cells were extracted with a total RNA extraction kit according to the manufacturer’s instructions (Invitrogen). All data were normalized to the expression of 18S ribosomal RNA. The primer sequences used in this study were as follows: 

PGC1α, 5’-ATCTACTGCCTGGGGACCTT-3’, 5’-ATGTGTCGCCTTCTTGCTCT-3’; 

NRF1, 5’-GCCGTCGGAGCACTTACT-3’, 5’-CTGTTCCAATGTCACCACC-3’; 

Tfam, 5’-CGCAGCACCTTTGGAGAA-3’, 5’-CCCGACCTGTGGAATACTT-3’; and 

18S rRNA, 5’-AGGGGAGAGCGGGTAAGAGA-3’, 5’-GGACAGGACTAGGCGGAACA-3’.

### 2.4. Detection of Reactive Oxygen Species (ROS) and ATP Content

Mitochondrial superoxide was studied using MitoSox Red (Molecular Probes, Eugene, OR, USA), according to the manufacturer’s instructions. Briefly, C2C12 myotubes were incubated with 5 μM MitoSox Red for 30 min at 37 °C, protected from light. Cells were rinsed with warm PBS and then observed under a fluorescence microscope.

MitoTracker Red CM-H2XRos (Molecular Probes) was used to determine intracellular ROS generation. C2C12 myotubes were then incubated with 100 nM MitoTracker Red CM-H2XRos for 30 min at 37 °C. At the end of incubation, the cells were washed and then lysed with 0.5% Triton X-100. Cell lysates were centrifuged, and the supernatants were measured with a fluorescence spectrometer (GloMax Discovery, Promega, Sunnyvale, CA, USA) at an excitation wavelength of 515 nm and an emission wavelength of 535 nm. Optical density values were normalized to the protein levels [22]. The ATP content was determined by a bioluminescence assay (ATP Determination Kit, Molecular Probes) according to the manufacturer’s instructions. The assay utilizes the enzyme luciferase to catalyze the formation of light from ATP and luciferin [23]. The cell lysates from C2C12 myotubes reacted with luciferin-luciferase reagent, and luminescence was measured using a luminometer (GloMax Discovery).

### 2.5. Determination of Mitochondrial Membrane Potential (MMP)

MMP was assessed using the JC-1 probe as per the manufacturer’s instructions. Briefly, C2C12 myotubes were cultured in culture plates with different treatments, incubated with 10 μg/mL of JC-1 (Molecular Probes) for 30 min at 37 °C, and then analyzed with a fluorescence spectrometer (GloMax Discovery). The fluorescence ratio (590–530 nm) was used for quantitative analysis [24]. For fluorescence microscopy, red fluorescence represents J-aggregates, whereas green fluorescence represents the monomeric form of JC-1. Images were captured and processed with a Nikon ECLIPSE TE-2000U microscope and NIS-Elements F software.

### 2.6. Peroxisome Proliferator-Activated Receptor-γ Coactivator 1α (PGC1α) Luciferase Assay

The PGC1α luciferase assay was performed as previously described [25]. The 5’ flanking sequence of the mouse PGC1α gene was amplified by PCR and subcloned into the pGL3basic reporter gene vector. In the plasmid, the region between base pairs +78 and −2533 with respect to the transcriptional start site is referred to as the 2 kb promoter. Briefly, C2C12 cells were seeded in 12-well plates at a density of 4 × 10^4^ cells per well. Twenty-four hours after seeding, cells were transfected with 100 ng of a luciferase plasmid encoding the full 2 kb promoter region of PGC1α (pPGC1α-Luc, Addgene, Cambridge, MA, USA), using Lipofectamine 2000. Twelve hours later, transfected cells were transferred into GM, harvested, and firefly luciferase activity was observed using a luminometer with the Luciferase Reporter Assay System (Promega). Experiments were performed in triplicate and repeated at least three times independently.

### 2.7. Preparation of BST204 (Fermented Ginseng Extract)

BST204 was provided by the Green Cross WellBeing, Co, Ltd. (Seongnam, Korea), and it was prepared with a patented technology and earlier study [17]. Briefly, the harvested ginseng was extracted with ethanol repeatedly followed by reaction with an enzyme containing ginsenoside-β-glucosidase. After acid hydrolysis of the residue, the reactant was purified with HP-20 resin, followed by washing out with distilled water and finally 95% ethanol. The 95% ethanol fraction was concentrated and designated BST204. As a result of analysis by HPLC-UV, the ginsenoside content of BST204 was 10.95% of Rg3 and 7.22% of Rh2. The NMR data and structure of BST204 are presented in previous studies. [26].

### 2.8. Statistical Analysis

The experiments were performed independently at least three times. The Student’s t-test with two-tailed paired comparison was used to assess the significance of the difference between two mean values; * *p* < 0.05 and ** *p* < 0.01 were considered to be statistically significant. The statistical significance was analyzed using SPSS (12.0 version: SPSS, Chicago, IL, USA).

## 3. Results and Discussion

### 3.1. BST204 Induces Myotube Hypertrophy Through Akt Activation

To investigate the effect of BST204 on myotube hypertrophy, C2C12 cells were induced to differentiate for two days (D2), followed by the treatment with 25 μg/mL BST204 for two days (D4). These cells were then subjected to immunostaining for a muscle differentiation marker myosin heavy chain (MHC) to assess myotube formation. Treatment with BST204 enhanced myotube formation, as evidenced by a higher proportion of larger myotubes containing more than 10 nuclei, relative to the vehicle-treated cells (Figure 1a,b). Measurement of myotube diameters showed that BST204 treatment increased the myotube thickness by about 1.5-fold (Figure 1c). To further examine the effect of BST204 on the expression of muscle-specific proteins and involved signaling pathways, C2C12 cells were cultured under the same experimental conditions as mentioned above and subjected to western blot analysis. Myotubes-treated with BST204 displayed higher levels of MHC and myogenin, compared to vehicle-treated myotubes (Figure 1d). In addition, BST204 dramatically upregulated Akt activation, as evidenced by greatly enhanced levels of the phosphorylated forms (p-Akt) (Figure 1d). These data indicate that BST204 induces myotube hypertrophy, which is dependent on the activation of Akt signaling to enhance myoblast differentiation.

### 3.2. BST204 Ameliorates DEX-Induced Myotube Atrophy Through Akt/mTOR Signaling 

The synthetic glucocorticoid DEX causes a distinct atrophic phenotype with a reduction in myotube diameter, and an increase in the levels of muscle-specific ubiquitin ligases [20,27]. To examine whether BST204 can affect the atrophy-induced upregulation of Atrogin-1 and MuRF1, we treated C2C12 myotube cultures with DEX. C2C12 cells were induced to differentiate for three days (D3), followed by treatment with DEX along with DMSO or BST204 for one day (D4). These myotubes were then subjected to MHC immunostaining. BST204 treatment enhanced the formation of large multinucleated myotubes relative to vehicle-treated cells (Figure 2a). As expected, DEX treatment blocked myotube formation, whereas BST204 treatment in DEX-treated C2C12 cells restored myotube formation, as evidenced by the presence of larger, multinucleated myotubes. Quantification of myotube diameter indicated that treatment with DEX induced a reduction in myotube diameter, which was partially recovered by treatment with BST204, and became comparable to the myotube diameter in the vehicle control (Figure 2b).

Next, we determined the effect of BST204 on the expression levels of MHC, myogenin, and E3 ubiquitin ligases in myotubes. As shown in Figure 2c, treatment with BST204 increased the expression levels of MHC and myogenin, while treatment with DEX decreased these proteins. Treatment with BST204 in DEX-treated myotubes restored the expression of MHC and myogenin and markedly decreased the expression of Atrogin-1 and MuRF1 compared to that in the DEX-treated control (Figure 2c). In addition, treatment with BST204 dramatically increased the levels of p-Akt and its downstream substrates, mTOR and p70S6K (Figure 2d). DEX-treated myotubes had decreased levels of active Akt, mTOR, and p70S6K, while BST204 treatment partially recovered the levels of the phosphorylated proteins in DEX-treated myotubes and brought them to levels comparable to those in the control myotubes (Figure 2d). The dose-dependence assessment indicated that DEX-treated myotubes had decreased levels of MHC, myogenin, and p-Akt, and cotreatment with BST204 elevated the levels of these muscle-specific proteins and the phosphorylated Akt in a dose-dependent manner (Figure 2e). In addition, BST204 treatment in DEX-treated myotubes also decreased the expression of Atrogin-1 and MuRF1 in a dose-dependent manner, compared to DEX-treated controls (Figure 2e). FoxO3 has been associated with muscle atrophy through the induction of MuRF1 and atrogin-1 [28]. FoxO3 coordinates the activation of protein breakdown via autophagic and proteasomal pathways in atrophic muscle cells [29]. Akt phosphorylates and inhibits FoxOs, thereby promoting cell survival [13]. Inhibition of FoxO by activation of Akt/mTOR signaling attenuates protein degradation and increases protein synthesis, leading to muscle regeneration and hypertrophy [14]. Taken together, these results suggest that BST204 prevents myotube atrophy induced by DEX through inhibition of muscle-specific ubiquitin ligases, mediated by activation of Akt/mTOR signaling.

### 3.3. BST204 Improves Mitochondrial Function in DEX-Treated Myotubes

To explore the effects of BST204 on mitochondrial function during myotube atrophy, C2C12 cells were induced to differentiate in DM for three days (D3) and treated with DEX along with the vehicle or BST204 for an additional one day (D4). We assessed mitochondrial ROS levels using MitoSox staining in C2C12 myotubes. Cotreatment with BST204 inhibited the elevated production of mitochondrial ROS induced by DEX (Figure 3a). Mitotracker Red CM-H2XRos is an oxidant sensing and cell-permeant fluorescent probe that is known to be mitochondrial-specific [22]. Once intracellular, it is oxidized by ROS to a fluorescent mitochondrion-sensitive probe and sequestered in this organelle. As a result of intracellular ROS production analysis, a significant increase in ROS production was observed after exposure to DEX in C2C12 myotubes, and treatment with BST204 significantly inhibited intracellular ROS production (Figure 3b). 

Decreased MMP represents a reduced ability of mitochondrial ATP synthesis to meet cellular energy needs [30]. Treatment with DEX led to a marked decrease in MMP, which was partially rescued by BST204 treatment (Figure 3c). To confirm the effect of BST204 on MMP, we performed the JC-1 fluorescence assay. As shown in Figure 3d, treatment with DEX increased green fluorescence due to the loss of MMP, whereas cotreatment with BST204 decreased the green fluorescence and thus increased the red/green fluorescence ratio. Consistent with the MMP results, treatment with BST204 restored ATP content in DEX-treated myotubes to levels comparable to those in control myotubes (Figure 3e). Disturbance in the mitochondrial network and function is associated with muscle weakness and FoxO3-dependent muscle atrophy, leading to decreased muscle function [31]. These results indicate that BST204 protects myotubes against oxidative damage and mitochondrial dysfunction in DEX-induced atrophic conditions.

### 3.4. BST204 Improves the Activity and Expression of PGC1α in DEX-Treated Myotubes

PGC-1α functions as a transcriptional coactivator and regulates the expression of genes associated with exercise, including genes involved in mitochondrial biogenesis, stimulation of fatty acid oxidation, angiogenesis, and resistance to muscle atrophy [32].

Elevated PGC1α levels have been reported to be beneficial in the prevention of age-related metabolic diseases and muscle atrophy [33]. We investigated whether BST204 modulates the activity of PGC1α in DEX-treated myotubes. C2C12 cells were transiently transfected with a PGC1α-responsive luciferase reporter and 24 h later, these cells were induced to differentiate in the presence of DEX along with the vehicle or BST204 for one day (D2), followed by a luciferase assay. As a result, treatment of BST204 partially arrested the decrease of DEX-induced PGC1α-luciferase activity (Figure 4a). For further investigation, we examined the expression levels of PGC1α and mitochondrial transcription factors. C2C12 cells were induced to differentiate in DM for three days (D3), followed by treatment with DEX along with vehicle or BST204 for one day (D4). Cell lysates were subjected to immunoblotting. Treatment with DEX dramatically decreased the levels of PGC1α, nuclear respiratory factor-1 (NRF1), and mitochondrial transcription factor A (Tfam), while cotreatment with BST204 partially restored the expression of these proteins in DEX-treated myotubes (Figure 4b). Consistent with the immunoblotting results, qRT-PCR analysis revealed that BST204 partially restored the transcription levels of PGC1α, NRF1, and Tfam in DEX-treated myotubes (Figure 4c). PGC1α with NRF1 promotes the expression of Tfam [34] and protects skeletal muscles from atrophy by suppressing the expression of FoxO3-mediated atrophy-specific gene [33]. It has been reported that increased PGC1α expression is sufficient to inhibit FoxO3-induced muscle fiber atrophy, and transgenic muscle-specific PGC1α overexpression protects against denervation- and fasting-induced atrophy associated with a reduction in the expression of Atrogin-1 and MuRF1 [33]. The activation of this PGC1α-NRF1-Tfam pathway leads to synthesis of mitochondrial DNA and proteins, and generation of new mitochondria [35]. Age-related changes in ROS metabolism can cause damage to mitochondrial DNA, but mainly due to an imbalance between mitochondrial biogenesis and mitophagy [36]. The modulation of PGC1α levels in skeletal muscle provides an avenue for the prevention and treatment of muscle aging and related metabolic diseases. Taken together, these results indicate that the expression and activity of PGC1α likely contributes to the enhanced mitochondrial function induced by BST204 treatment.

## 4. Conclusions

This study provides a mechanistic insight into the effect of BST204 on hypertrophy and myotube atrophy. Our data suggest that the effect of BST204 is mediated through activation of hypertophic signaling as well as suppression of atrophy pathways. This finding, for the first time, provides critical clues for understanding how BST204 protects against DEX-induced myotube atrophy at the molecular level, suggesting that Rh2 and Rg3 might be potential herbal medicinal products to intervene muscle weakness and atrophy, including cancer cachexia. C2C12 cells are a very useful model for doing muscle-related studies, but there are limitations and DEX-induced atrophy does not reflect many of the processes associated with muscle atrophy occurring in various diseases, so it should be validated in more complex experimental models.

## Figures and Tables

**Figure 1 ijerph-18-02367-f001:**
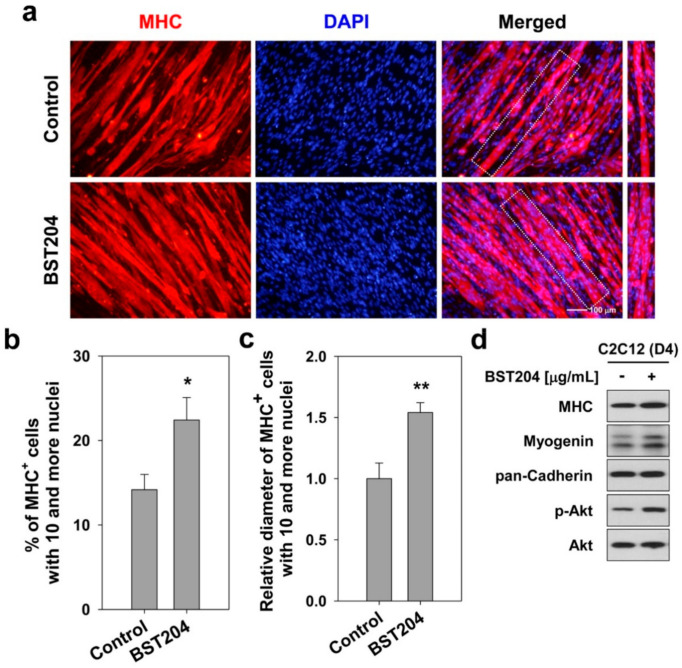
BST204 promotes Akt-dependent myotube hypertrophy. (**a**) C2C12 cells were induced to differentiate for 2 days in DM, and then treated with 25 μg/mL BST204 for 2 days. These myotubes were then subjected to immunostaining for MHC expression (red) and DAPI to visualize nuclei (blue). (**b**) The MHC-positive myocytes shown in panel **a** were quantified as 10 or more nuclei per myotube. * *p* < 0.05 compared with vehicle-treated group. (**c**) Quantification of average myotube diameter from data shown in panel **a**. ** *p* < 0.01 compared with vehicle-treated group. (**d**) Cell lysates from panel **a** were subjected to immunoblotting analysis. This experiment was repeated twice with similar results.

**Figure 2 ijerph-18-02367-f002:**
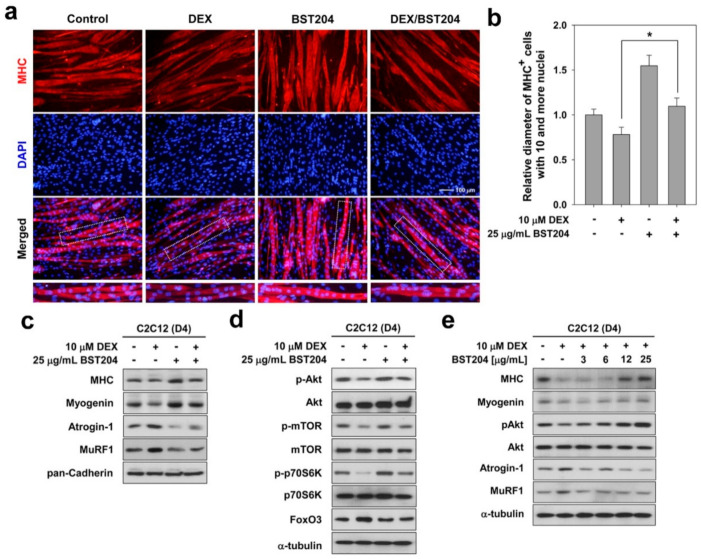
BST204 prevents dexamethasone (DEX)-induced myotube atrophy through Akt activation. (**a**) C2C12 cells were induced to differentiate in DM for 3 days, followed by the treatment with 10 μM DEX along with vehicle or 25 μg/mL BST204 for 1 day. These myotubes were then subjected to immunostaining for MHC expression (red) and DAPI to visualize nuclei (blue). (**b**) Quantification of average myotube diameter from data shown in panel **a**. Significant difference from DEX-treated group, * *p* < 0.05. (**c**) C2C12 cells were induced to differentiate in DM for 3 days, followed by the treatment with 10 μM DEX along with vehicle or 25 μg/mL BST204 for 1 day. Cell lysates were subjected to immunoblotting analysis. (**d**) Cell lysates from panel **c** were subjected to immunoblotting analysis. (**e**) C2C12 cells were induced to differentiate in DM for 3 days, followed by the treatment with 10 μM DEX along with vehicle or the indicated concentration of BST204 for 1 day. Cell lysates were subjected to immunoblotting analysis. All immunoblotting experiments were repeated three times with similar results.

**Figure 3 ijerph-18-02367-f003:**
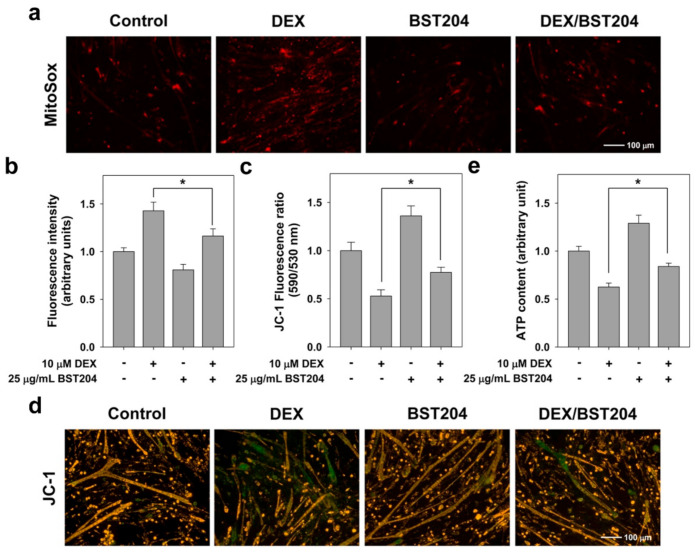
BST204 impedes the production of mitochondrial ROS induced by DEX. C2C12 cells were induced to differentiate in DM for 3 days, and then treated with 10 μM DEX along with vehicle or 25 μg/mL BST204 for 1 day. (**a**) Mitochondrial ROS production analysis. Differentiated C2C12 myotubes were stained with MitoSox Red and observed by fluorescence microscope. (**b**) intracellular ROS production analysis using Mitotracker Red CM-H2XRos. Fluorescence intensity is given as arbitrary units and values presented as means ± SD (*n* = 4). Significant difference from DEX-treated group, * *p* < 0.05. (**c**) Mitochondrial membrane potential (MMP) analysis. Fluorescence ratio (590 nm to 530 nm) was used for quantitative analysis. Significant difference from DEX-treated group, * *p* < 0.05. (**d**) Analysis for MMP by JC-1 staining. (**e**) The determination of ATP content. The total ATP content was measured using ATP Determination Kit. Values presented are as means ± SD (*n* = 5). * *p* < 0.05 compared with DEX-treated group.

**Figure 4 ijerph-18-02367-f004:**
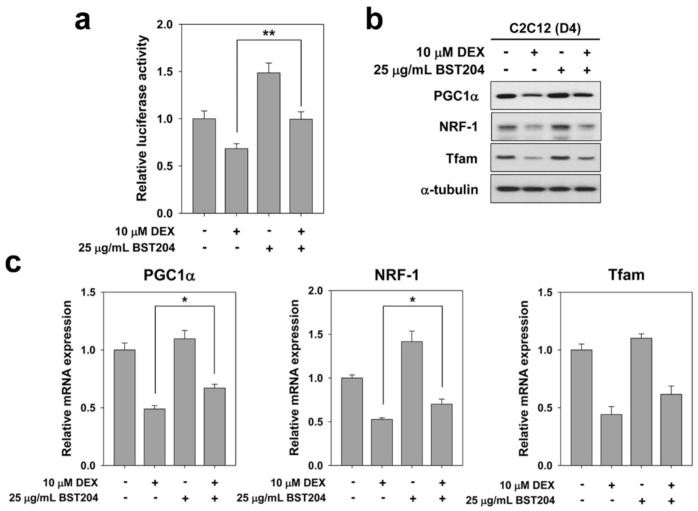
BST204 upregulates the PGC1α activity in DEX-induced atrophic myotubes. (**a**) C2C12 cells were transiently transfected with a PGC1α-responsive luciferase reporter. After 24 h, these cells were induced to differentiate in the presence of 10 μM DEX, along with vehicle or 25 μg/mL BST204 for an additional 1 day, followed by luciferase assay. ** *p* < 0.01 compared with DEX-treated group. (**b**) C2C12 cells were induced to differentiate in DM for 3 days, followed by the treatment with 10 μM DEX along with vehicle or 25 μg/mL BST204 for 1 day. These cell lysates were then subjected to immunoblotting analysis. The experiment was repeated three times with similar results. (**c**) Quantitative RT-PCR analysis for PGC1α, NRF-1 and Tfam in C2C12 cells treated with 10 μM DEX, along with vehicle or 25 μg/mL BST204 at D2. All values from control sample were set to 1.0. Data from three independent experiments were presented as the means ± SD. Significant difference from DEX-treated group, * *p* < 0.05.

## Data Availability

Data is contained within the article.

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
