# Peer review of "BST204 Protects Dexamethasone-Induced Myotube Atrophy through the Upregulation of Myotube Formation and Mitochondrial Function"

_ijerph, 2021, doi:10.3390/ijerph18052367_

Round 1
Reviewer 1 Report
The present manuscript presents new results and constitutes an advancement of the state-of-the-art, and I think it should be published. I detected some minor errors that I recommend authors to correct, before publishing:
-Line 14-15 . "inhibitory effect on...cancer" too much speculative sentence. Please rephrase.
-Methods 2.8 Statistical analysis - which software was used to perform statistics?
-Figure 1 and Figure 2: please add to the figure´s caption the statistic significance values (* and **)
Author Response
Response to Reviewer 1 Comments
The present manuscript presents new results and constitutes an advancement of the state-of-the-art, and I think it should be published. I detected some minor errors that I recommend authors to correct, before publishing:
We thank the reviewer for the encouraging comment.
Point 1: Line 14-15. "inhibitory effect on...cancer" too much speculative sentence. Please rephrase.
Response 1: As the reviewer suggested, we have corrected it in the revised manuscript and changed to “an inhibitory effect on lipopolysaccharide-induced inflammatory responses”.
Point 2: Methods 2.8 Statistical analysis - which software was used to perform statistics?
Response 2: We have used SPSS (version 12.0) to analyze statistical significance and described this in “2.8 Statistical analysis” of the revised manuscript.
Point 3: Figure 1 and Figure 2: please add to the figure´s caption the statistic significance values (* and **)
Response 3: As the reviewer suggested, we added the statistically significant values in the caption of Figure 1 and Figure 2.
Reviewer 2 Report
The experiments are appropriately designed and well described, there are only few aspects, which shall be improved:
- The validity of the model shall be briefly commented: C2C12 myotubes is a very useful and well described model for studying muscle differentiation and muscle cell physiology, however it has clear limitations (e.g. differentiation only to a certain stage). Also DEX-induced atrophy does not reflect many processes related to muscle atrophy occurring in various diseases. Thus, the results obtained with the applied approach are promising, but need to be validated in more complex experimental models. These aspects shall be shortly mentioned in the introduction and/or conclusions.
- MitoTracker Red CM-H2XRos is not an appropriate probe for mitochondrial ROS determination. According to the producer’s information, it is oxidized in the cytosol and then enters the mitochondria in membrane potential-dependent manner. The measurements performed with this probe are valid, but shall not be interpreted as “mitochondrial ROS production” but rather as general intracellular ROS production. This shall be corrected in the Methods and in Results sections. For quantification of mitochondrial ROS production, MitoSOX red probe is more appropriate, and the attempt for semiquantitative analysis can be made based on the obtained microscopic images.
- Statistical analysis shall be performed by comparing the treatment effects with the vehicle-treated controls. Comparison with DEX-treated cells makes sense only in case of DEX+BST group, comparing BST-only group with DEX-only group brings no valuable information. Thus, the statistical analysis need to be corrected.
- Figures 1, 3 and 4 are missing the information on the amount of independent repetitions of the immunoblotting experiments (it is stated only in the legend to Fig. 2)
- Line 75 – instead of reference 19, the reference containing the description of the method shall be given (ref. 19 refers to some further papers concerning the method’s description)
- Line 163 – is “participants’ t-test” the correct name of the used statistical test?
- Small typos – “alpha” is missing in lines 279 and 300
Author Response
Response to Reviewer 2 Comments
The experiments are appropriately designed and well described, there are only few aspects, which shall be improved:
We thank the reviewer for his/her positive comment.
Point 1: The validity of the model shall be briefly commented: C2C12 myotubes is a very useful and well described model for studying muscle differentiation and muscle cell physiology, however it has clear limitations (e.g. differentiation only to a certain stage). Also DEX-induced atrophy does not reflect many processes related to muscle atrophy occurring in various diseases. Thus, the results obtained with the applied approach are promising, but need to be validated in more complex experimental models. These aspects shall be shortly mentioned in the introduction and/or conclusions.
Response 1: We agree with the reviewers on this point, and at the conclusion of the revised manuscript we have added a sentence related to this: “C2C12 cells are a very useful model for doing muscle-related studies, but there are limitations and DEX-induced atrophy does not reflect many of the processes associated with muscle atrophy occurring in various diseases, so it should be validated in more complex experimental models.”
Point 2: MitoTracker Red CM-H2XRos is not an appropriate probe for mitochondrial ROS determination. According to the producer’s information, it is oxidized in the cytosol and then enters the mitochondria in membrane potential-dependent manner. The measurements performed with this probe are valid, but shall not be interpreted as “mitochondrial ROS production” but rather as general intracellular ROS production. This shall be corrected in the Methods and in Results sections. For quantification of mitochondrial ROS production, MitoSOX red probe is more appropriate, and the attempt for semiquantitative analysis can be made based on the obtained microscopic images.
Response 2: We thank the reviewer for pointing out this. As the reviewer suggested, we have corrected it in the revised manuscript and changed “mitochondrial ROS production” into “intracellular ROS production” lines at 118, 261, 263 and 270.
Point 3: Statistical analysis shall be performed by comparing the treatment effects with the vehicle-treated controls. Comparison with DEX-treated cells makes sense only in case of DEX+BST group, comparing BST-only group with DEX-only group brings no valuable information. Thus, the statistical analysis need to be corrected.
Response 3: We thank the reviewer for pointing this out. As the reviewer suggested, we have corrected the statistical analysis that was compared DEX-treated group with DEX-BST204-treated group. And we have adjusted Figure 2b, 3b, 3c, 3e, 4a and 4c in the revised manuscript.
Point 4: Figures 1, 3 and 4 are missing the information on the amount of independent repetitions of the immunoblotting experiments (it is stated only in the legend to Fig. 2)
Response 4: As the reviewer suggested, we stated the information about replicates of immunoblotting experiment in figure legend of the revised manuscript.
Point 5: Line 75 – instead of reference 19, the reference containing the description of the method shall be given (ref. 19 refers to some further papers concerning the method’s description)
Response 5: As the reviewer suggested, we changed the reference that is firstly described this experimental method.
Point 6: Line 163 – is “participants’ t-test” the correct name of the used statistical test?
Response 6: We used Student’s t-test with two-tailed paired comparison to analyze statistical significance and have corrected statistical test in the revised manuscript.
Point 7: Small typos – “alpha” is missing in lines 279 and 300
Response 7: We apologize and thank the reviewer for pointing out our mistake. We have corrected this mistake in the revised manuscript.
Reviewer 3 Report
In the manuscript entitled “BST204 Protects Dexamethasone-Induced Myotube Atrophy through the Upregulation of Myotube Formation and Mitochondrial Function”, the authors identified that BST204 could alleviates the myotube atrophic effect induced by DEX via the activation of protein kinase B/mammalian target of rapamycin (Akt/mTOR) signaling. Further, they revealed BST204 could protect mitochondrial function via decreasing mitochondrial ROS and upregulating the expression of PGC1α. This observation is interesting. This manuscript was well written and the figures were well presented. Regarding the manuscript, I have a few minor concerns:
1) Where does BST204 come from? Is it commercial or your lab made it? Please clarify in the ‘Materials and Methods’ part.
2) For Figure 1, how did you calculate the diameter of the muscle cells? Did you measure the transverse or longitudinal diameter? Please make it clear in the ‘Materials and Methods’ part.
3) In the text and figure legend of the ‘Results’ part, the authors used MitoSox to measure mitochondrial ROS production. However, the authors wrote and indicated using MitoTracker Red CM-H2XRos to measure ROS production, so the ‘method’ part and the ‘Results’ part are not consistent. The authors introduced the staining of muscle cells by using MitoSox in another part, so they should move that part to the ‘Detection of mitochondrial reactive oxygen species (ROS)’, and remove the staining of Mitotracker Red, for I did not see any results in the manuscript which used Mitotracker Red.
4) In figure 4, the authors observed increased expression of PG1Cα after the treatment of BST204. PG1Cα is a master regulator of mitochondrial biogenesis. Consistent with this notion, the data also showed increased Tfam for mitochondrial transcription. So did you see any changes of mitochondrial morphologies and numbers by using Mitotracker Staining? Now that the author already did Mitotracker staining, you can add your data here. And you can do some discussion on the role of BST204 in mitochondrial biogenesis.
5) The ‘α’ is missing on lines 279 and 300.
Author Response
Response to Reviewer 3 Comments
In the manuscript entitled “BST204 Protects Dexamethasone-Induced Myotube Atrophy through the Upregulation of Myotube Formation and Mitochondrial Function”, the authors identified that BST204 could alleviates the myotube atrophic effect induced by DEX via the activation of protein kinase B/mammalian target of rapamycin (Akt/mTOR) signaling. Further, they revealed BST204 could protect mitochondrial function via decreasing mitochondrial ROS and upregulating the expression of PGC1α. This observation is interesting. This manuscript was well written and the figures were well presented. Regarding the manuscript, I have a few minor concerns:
We thank the reviewer for his/her positive comment.
Point 1: Where does BST204 come from? Is it commercial or your lab made it? Please clarify in the ‘Materials and Methods’ part.
Response 1: BST204 was provided by Green Cross Wellbeing Co., Ltd., a cooperative research partner, and has been used for this study. And we described BST204 supplier in “Materials and Methods” part of the revised manuscript.
Point 2: For Figure 1, how did you calculate the diameter of the muscle cells? Did you measure the transverse or longitudinal diameter? Please make it clear in the ‘Materials and Methods’ part.
Response 2: We measured the transverse diameter of MHC-positive myotubes with 10 and more nuclei using Image J software. Average myotube diameter is presented as means determination of six fields ± 1 standard deviation (SD). We described this method in the part 2.2. of “Materials and Methods” of the revised manuscript.
Point 3: In the text and figure legend of the ‘Results’ part, the authors used MitoSox to measure mitochondrial ROS production. However, the authors wrote and indicated using MitoTracker Red CM-H2XRos to measure ROS production, so the ‘method’ part and the ‘Results’ part are not consistent. The authors introduced the staining of muscle cells by using MitoSox in another part, so they should move that part to the ‘Detection of mitochondrial reactive oxygen species (ROS)’, and remove the staining of Mitotracker Red, for I did not see any results in the manuscript which used Mitotracker Red.
Response 3: We apologize to cause your confusion. We performed both experiments of MitoSox and MitoTracker to measure mitochondrial and intracellular ROS production, respectively. In Figure 3a, we assessed mitochondrial ROS levels using MitoSox staining in C2C12 myotubes. And we carried out the analysis of intracellular ROS production using Mitotracker Red CM-H2XRos in Figure 3c. Although Mitotracker Red CM-H2XRos is known as mitochondrial specific oxidant sensing probe, it is oxidized by intracellular ROS and then moved the mitochondria in membrane potential-dependent manner.
As reviewer’s suggestion, the method of MitoSox was moved part 2.2 into part 2.4 of “Materials and Methods” and added together. We have adjusted the legends of Figure 3c in the revised manuscript.
Point 4: In figure 4, the authors observed increased expression of PG1Cα after the treatment of BST204. PG1Cα is a master regulator of mitochondrial biogenesis. Consistent with this notion, the data also showed increased Tfam for mitochondrial transcription. So did you see any changes of mitochondrial morphologies and numbers by using Mitotracker Staining? Now that the author already did Mitotracker staining, you can add your data here. And you can do some discussion on the role of BST204 in mitochondrial biogenesis.
Response 4: Unfortunately, we did not perform Mitotracker staining but have measured and analyzed a fluorescence. So, we could not check the changes of mitochondrial morphologies and numbers. We have briefly discussed on the role of BST204 in mitochondrial biogenesis in Part 3.4. of the revised manuscript.
Point 5: The ‘α’ is missing on lines 279 and 300.
Response 5: We apologize and thank the reviewer for pointing out our mistake. We have corrected this mistake in the revised manuscript.